# Impaired Height Growth Associated with Vitamin D Deficiency in Young Children from the Japan Environment and Children’s Study

**DOI:** 10.3390/nu14163325

**Published:** 2022-08-13

**Authors:** Shohei Kuraoka, Masako Oda, Hiroshi Mitsubuchi, Kimitoshi Nakamura, Takahiko Katoh

**Affiliations:** 1The South Kyushu Okinawa Unit Center, Faculty of Life Sciences, Kumamoto University, 1-1-1 Honjo, Kumamoto 860-8556, Japan; 2Department of Pediatrics, Faculty of Life Sciences, Kumamoto University, 1-1-1 Honjo, Kumamoto 860-8556, Japan; 3Department of Neonatology, Kumamoto University Hospital, 1-1-1 Honjo, Kumamoto 860-8556, Japan; 4Department of Public Health, Faculty of Life Sciences, Kumamoto University, 1-1-1 Honjo, Kumamoto 860-8556, Japan

**Keywords:** vitamin D, height growth, sun exposure, child health

## Abstract

Vitamin D is essential for calcium absorption and bone homeostasis. Although short-stature children were reported to have low vitamin D concentrations, there is no clear evidence of a link between vitamin D and height growth in young children not limited to those with short stature. We collected height and weight data at 2 and 4 years of age, serum vitamin D concentrations at 4 years, and questionnaire results on sun exposure from the Japan Environment and Children’s Study (JECS). We then analyzed the relationship between vitamin D deficiency and height growth. We also analyzed the correlation between serum vitamin D concentration and sun exposure. Overall, 3624 participants from JECS were analyzed. We identified cases of subclinical vitamin D deficiency and insufficiency. We further found that definitive vitamin D deficiency (<10 ng/mL) impaired height growth by 0.6 cm per year even in young children not limited to those with short stature. Furthermore, we clarified that children with vitamin D deficiency had reduced outdoor activity, especially during winter. In children with either short or normal stature, definitive vitamin D deficiency was associated with height growth decline, and reduction in outdoor activity, especially during winter, was a risk factor for vitamin D deficiency.

## 1. Introduction

Vitamin D is the precursor of 25-hydroxyvitamin D [25(OH)D], and 25(OH)D is converted to 1α-25-dihydroxyvitamin D, which regulates calcium absorption in the intestine. 25(OH)D has two forms: ergocalciferol [25(OH)D2] and cholecalciferol [25(OH)D3]. 25(OH)D3 is synthesized in the skin upon exposure to sunlight and is the dominant form in humans [1]. 25(OH)D3 has a half-life of approximately 15 days, while 1α-25-dihydroxyvitamin D3, the hormonal form, has a half-life of approximately 15 h [2]. Therefore, 25(OH)D3 is measured as a biomarker for vitamin D in circulation. Inadequate vitamin D can impair bone homeostasis and cause rickets in children as well as osteomalacia in adults [3,4,5]. Even subclinical vitamin D deficiency can be followed by increases in osteoporosis and fractures [5,6]. During the last decade, vitamin D supplementation has been encouraged in many countries [7,8], but not in Japan. To date, vitamin D has been reported to be strongly associated with not only skeletal health, but also immunity, cardiovascular disease, diabetes, and several cancers [3,9,10,11,12,13,14,15,16]. Although vitamin D was found to be low in children with short stature [17], there is no clear evidence of a link between child growth and vitamin D in a standard population.

We performed the Japan Environment and Children’s Study (JECS), a large-scale nationwide birth cohort study [18,19]. The JECS was designed to analyze the effects of various environmental factors on children’s health and development. The JECS Main Study, which enrolled 100,000 children and their parents, evaluated the effects of environmental factors using self-administered questionnaires. The JECS also included a Sub-Cohort Study, involving randomly selected participants from the JECS, that aimed to accumulate detailed data on developmental, environmental, and endocrinal findings [20]. Using the data from the Sub-Cohort Study in the present study, we investigated the effects of vitamin D on height growth in children with normal stature. We detected many children with subclinical vitamin D deficiency and insufficiency in a standard population. We further found that vitamin D deficiency impaired height growth in young children not limited to those with short stature.

## 2. Materials and Methods

### 2.1. Study Participants

In the JECS, participating mothers were recruited across 15 Regional Centers from January 2011 to March 2014 [18]. While the JECS Main Study recruited around 100,000 pregnant women, the Sub-Cohort Study started recruiting in April 2013 with the aim of enrolling 5000 participants. The recruitment details for the Sub-Cohort study are described in a previous report [20]. In the present study, we targeted 3624 children born at term (pregnancy period >37 weeks) without any complications (such as congenital heart disease) to the participants for whom accurate data obtained at 2 and 4 years of age were available.

### 2.2. Data Collection

In accordance with the overall data collection schedule in the Sub-Cohort Study, medical examinations were conducted by pediatricians when the participants reached 2 and 4 years of age (period of survey at 2 years of age: April 2015 to January 2017; period of survey at 4 years of age: April 2017 to February 2019). These medical examinations included measurements of height (SECA 213 Stadiometer; SECA, Hamburg, Germany) and weight (Tanita WB-260A Digital Body Scale; Tanita, Tokyo, Japan) and blood sample collection. To relieve pain and distress during the blood sample collection as much as possible, pediatricians and healthcare providers who had received specific training sampled the blood from the participants with great care [21]. Serum 25(OH)D3 was measured as one of the variables in the blood samples from the medical examinations by liquid chromatography-tandem mass spectrometry (LSI Medience Corporation, Tokyo, Japan). The system was calibrated with 6PLUS1 Multilevel Serum Calibrator Set 25-OH-Vitamin D3/D2. The sample testing was validated through Accuracy-Based Vitamin D Survey (ABVD). The intra-assay CV for 25(OH)D3 was L: 9.9% (17.2 ng/mL) and H: 9.1% (44.2 ng/mL). The inter-assay CV for 25(OH)D3 was L: 4.7% (15.7 ng/mL) and H: 5.2% (39.3 ng/mL). The data described above were extracted from the following datasets: jecs-qa-20210401, jecs-ta-20190930.

From the questionnaire survey at 4 years of age conducted in the same way as in the JECS Main Study schedule (http://www.env.go.jp/chemi/ceh/, accessed on 13 March 2022), we collected data on time spent playing outside during the daytime (9:00 a.m. to 5:00 p.m.) in two seasons (summer: May to September; winter: November to February), frequency of wearing a hat, and frequency of using sunscreen in summer. These questionnaire data were extracted from the following dataset: jecs-qa-20210401.

### 2.3. Calculation of Standard Scores and Height Growth

To calculate the standard deviation scores (SDSs) for height, weight, and body mass index (BMI), we used the Excel-based Clinical Tools for Growth Evaluation of Children provided by the Japanese Society for Pediatric Endocrinology (http://jspe.umin.jp/medical/chart_dl.html, accessed on 2 December 2021). The difference in height measured at 2 and 4 years of age was corrected to the value per 12 months to obtain the annual height growth.

### 2.4. Statistical Analysis

Data are presented as mean with standard deviation (SD) or 95% confidence interval (95% CI). Statistical analyses were performed by the Tukey–Kramer test for differences among multiple groups with parametric data, and by the Steel–Dwass test for differences among multiple groups with non-parametric data. Differences with values of *p* < 0.05 were considered statistically significant.

## 3. Results

Among the 5015 participants selected for the Sub-Cohort Study in the JECS, 3947 participants were under investigation at both 2 and 4 years of age. To evaluate the relationship between growth and vitamin D concentration accurately, we excluded children with one or more complications. We further excluded participants who were born prematurely (pregnancy period <37 weeks) to avoid concerns on the effects of prematurity. Finally, we analyzed data for 3624 participants in the present study (Figure 1). The characteristics of these participants are shown in Appendix A (online only). The sex ratio, pregnancy period, birth weight, and birth height were similar to the overall profiles in the JECS and the original Sub-Cohort Study [18,20]. Because the height at 2 years of age was shorter (mean SDS: −0.32) than the standard height, the BMI at 2 years of age was slightly high (mean SDS: 0.49). However, the SDSs for height, weight, and BMI at 4 years of age were close to zero, indicating that the participants surveyed in this study were representative of the overall population. At 4 years of age, the number of children with short stature, defined as height SDS below −2, was 49, accounting for 1.35% of the total cohort.

The concentrations of 25(OH)D3 in serum, known to be its dominant location in humans, were measured using the blood samples collected at 4 years of age and presented a parametric distribution (Figure 2A). The median serum 25(OH)D3 concentration was 25.6 ng/mL. Surprisingly, a concentration of <20 ng/mL, defined as vitamin D insufficiency in the Global Consensus [22], was found in 23.1% of the total cohort, and a concentration of <10 ng/mL, defined as vitamin D deficiency, was found in 1.1%. We also checked the monthly average concentrations because the survey was conducted throughout the year (April 2017 to February 2019) and confirmed that the 25(OH)D3 concentration fluctuated seasonally (Figure 2B), consistent with previous reports [6,23,24,25]. The highest monthly mean serum 25(OH)D3 concentration was 31.28 (95% CI: 30.55–32.00) ng/mL in September and the lowest was 18.30 (95% CI: 17.33–19.27) ng/mL in February. Thus, we revealed that subclinical vitamin D deficiency and insufficiency may occur in Japanese children. We further confirmed that the serum 25(OH)D3 concentration declined by >10 ng/mL in winter (January to March) compared with that in summer (August and September).

To examine the concentration-dependent effect of 25(OH)D3 on childhood growth, we classified the participants into seven groups based on their serum 25(OH)D3 concentrations (<10, *n* = 40; ≥10 to <15, *n* = 254; ≥15 to <20, *n* = 524; ≥20 to <25, *n* = 913; ≥25 to <30, *n* = 925; ≥30 to <40, *n* = 846; ≥40, *n* = 122) and aggregated the height, weight, and BMI for each group (Table 1). There were no differences in pregnancy period, birth weight, and birth height among the seven groups. Despite the lack of differences in body weight or BMI, height tended to be shorter in the lowest 25(OH)D3 group (<10 ng/mL) at 2 years of age, although the difference was not significant. This tendency became marked at 4 years of age, and a significant difference was confirmed in the lowest 25(OH)D3 group (<10 ng/mL) compared with the findings in the two highest groups (≥30 to <40, *p* = 0.016; ≥40, *p* = 0.027). We further calculated the height growth per year (Δheight). The results showed that Δheight was extremely low in the lowest 25(OH)D3 group (<10 ng/mL) and a significant difference was even found in the comparison with the second lowest group (Figure 3). Compared with the groups with sufficient 25(OH)D3 (≥20 ng/mL), Δheight in the lowest group was as low as 0.6 cm per year. Meanwhile, there were no clear differences in body weight or BMI among the seven groups (Table 1). Furthermore, there were no children with emaciation, defined as BMI SDS below −2, in the lowest group. These results suggest that the severe decline in serum 25(OH)D3 concentration was not caused by undernutrition. Therefore, the results showed that 25(OH)D3 deficiency (<10 ng/mL) impaired height growth in young children, regardless of emaciation or undernutrition. In the lowest group, there were only two children (5%) with short stature, indicating that 25(OH)D3 deficiency is associated with impaired height growth in a standard population not limited to those with short stature.

Next, we investigated the frequency of exposure to sunlight to determine the cause of the low 25(OH)D3 concentrations. Using the questionnaire sent out at 4 years of age, we conducted an analysis on the time spent playing outside in both summer and winter, and the frequencies of wearing a hat and using sunscreen. The results are shown in Appendix A (online only). In all groups, the frequency of wearing a hat in summer was as high as 80% or more, for which there was no clear difference among the groups. Similarly, the frequency of using sunscreen in summer was 20–30%, for which there was no clear difference among the groups. However, when the total time spent playing outside was determined, it tended to be inversely proportional to the serum 25(OH)D3 concentration. About 30% of children in the lowest 25(OH)D3 group spent <1 h playing outside in the summer, compared with about 20% in the other groups (Figure 4 and Appendix A). The difference in time spent playing outside among the groups increased in winter, and a significant difference was found between the groups with 25(OH)D3 < 25 ng/mL and the groups with 25(OH)D3 > 30 ng/mL (Figure 4B). In the lowest group, >60% of participants answered that they spent <1 h playing outside. These results are consistent with previous findings showing that time spent with exposure to sunlight affects the serum vitamin D concentration [1,26]. Interestingly, the results suggested that differences in sun exposure during winter may have a greater effect on the decline in 25(OH)D3.

## 4. Discussion

We analyzed data obtained from the Sub-Cohort Study of the JECS to examine the effect of vitamin D levels on childhood growth. Although the study excluded children born prematurely and children with complications, we were able to retain enough participants to achieve sufficient statistical power in our analyses. The analyses revealed that subclinical vitamin D deficiency is reasonably widespread in Japanese children. More than one in five children did not achieve sufficient vitamin D levels, and a concentration of ≤10 ng/mL, considered to reflect definite vitamin D deficiency, was confirmed at a rate of 1% or more. Furthermore, the height growth in this group with extremely low vitamin D was found to be about 0.6 cm per year less than that in the group with sufficient vitamin D. Although some reports of a relationship between short stature and low vitamin D have been published, the group with the lowest vitamin D only contained two children with height SDS below −2 among 40 children, indicating that the effect of vitamin D deficiency was not limited to children with short stature. Therefore, this is the first study to prove that vitamin D deficiency is associated with impaired height growth in a standard pediatric population. When the serum 25(OH)D3 concentration was low but maintained above 10 ng/mL, height growth did not differ significantly from that in the sufficient group, suggesting that 25(OH)D3 of 10 ng/mL is the threshold for this phenotype. Considering that vitamin D promotes calcium absorption in the intestinal tract and is essential for maintaining the blood calcium concentration and normal bone metabolism, the present results were reasonable. However, because the present survey did not measure the blood calcium level or bone density, it is difficult to determine whether the decrease in height growth due to vitamin D deficiency was mediated by a decline in the blood calcium level. At least, because the participants were completely asymptomatic, it is unlikely that they had severe hypocalcemia. In the future, further analyses are expected to clarify the relationship among vitamin D, calcium, and height growth.

The groups with low vitamin D tended to spend less time playing outside, and this difference compared with the groups with sufficient vitamin D was particularly noticeable in winter. Considering the seasonal fluctuations in vitamin D, the results suggested that excessive time spent indoors in winter, with decreased exposure to sunlight, may cause a severe decline in vitamin D. It was previously reported that the monthly height growth rate for children in Japan fluctuates in proportion to the hours of sunshine [27], and the vitamin D concentration seems to correlate with the fluctuations in height growth. Since this study recruited participants from throughout the Japanese archipelago from latitude 26° N to latitude 43° N, latitude may have an effect. However, in the northernmost region of the country, the hours of winter sunshine number about 110 per month, while in the southernmost region they number about 98, which are not so different. Therefore, decreased outdoor activity due to the winter climate, rather than the number of hours of sunshine, may be the main factor in reduced exposure to sunlight at each latitude. The difference in height growth (0.6 cm per year) in the present study may have mainly been caused by the impaired height growth in winter, a season associated with a risk of vitamin D decline.

This study had several limitations. First, the data analyzed in the study did not include the calcium concentrations, as mentioned above, and other growth-related biochemical data (such as growth hormone and thyroid hormone concentrations), which could support considerations on the mechanism behind the vitamin D deficiency induced decline in height growth. Second, we did not carry out sufficient medical examinations and tests (such as X-rays) to determine whether the children with vitamin D deficiency had rickets. Therefore, the possibility of rickets could not be fully ruled out, and the impaired height growth may have been caused by bowed legs or knock knees. Third, 25(OH)D3 was not measured over time during the 2-year observation period in this study. Considering the length of the half-life of 25(OH)D3, the values measured in the study may represent habits from about 2–3 weeks before the time of blood sample collection. Since the height growth per year was calculated from the average value for the observation period of 2 years, the influence of seasonal fluctuations depending on the measurement date may have been suppressed. However, frequent measurement of 25(OH)D3 and height growth will reveal the details of the underlying mechanism.

The present study confirmed the value of the JECS, a large-scale epidemiological study. A difference of 0.6 cm per year is not small for young children, but it may not be easy to prove the significance of this difference unless a large study is performed. In the survey conducted for the JECS, the same instrument for height measurement was used at all facilities and the same measurement method was used. The advantage of being able to track large data collected in a uniform manner in chronological order was essential for the present study. In the JECS, the plan is to continue the study until the children are 13 years of age and further clarify the long-term effects of lowered vitamin D.

## 5. Conclusions

We revealed the potential occurrence of vitamin D deficiency in 4-year-old children by analyzing data in the JECS, a nationwide survey. We also proved that 25(OH)D3 deficiency is associated with impaired height growth in children. To the best of our knowledge, this is the first study to provide evidence of a relationship between vitamin D and height growth in a standard population not limited to those with short stature. The present results confirm that serum vitamin D is essential for children with poor height growth, even those without short stature. The results further suggest that reduced sun exposure in winter is a risk factor for a severe decline in vitamin D. Our findings encourage adequate sun exposure time, even in winter, to maintain normal height growth in young children. In the future, intervention studies such as those involving the use of vitamin D supplements and lifestyle changes to promote sunlight exposure are expected to elucidate the mechanism by which vitamin D deficiency suppresses height growth in children.

## Figures and Tables

**Figure 1 nutrients-14-03325-f001:**
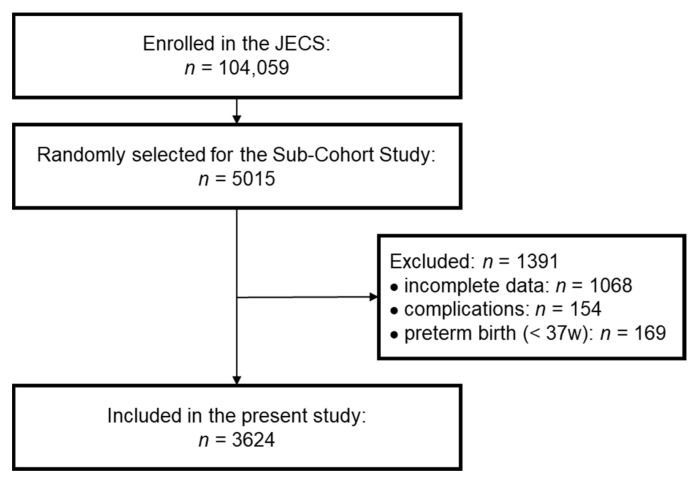
Selection flow diagram. Participants in the Sub-Cohort Study were randomly selected from the Japan Environment and Children’s Study (JECS). Finally, participants who did not meet the exclusion criteria (incomplete data, complications, preterm birth) were included in the present study.

**Figure 2 nutrients-14-03325-f002:**
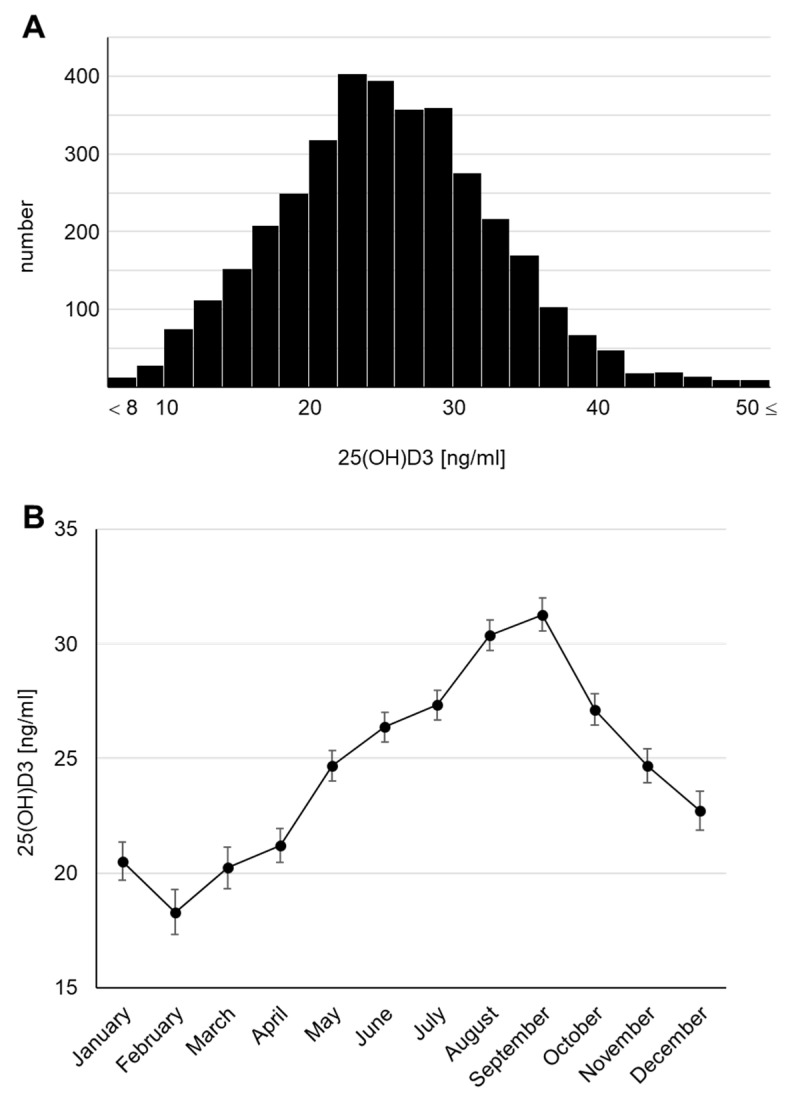
Serum 25(OH)D3 concentrations at 4 years of age (**A**) Histogram. (**B**) Monthly transitions.

**Figure 3 nutrients-14-03325-f003:**
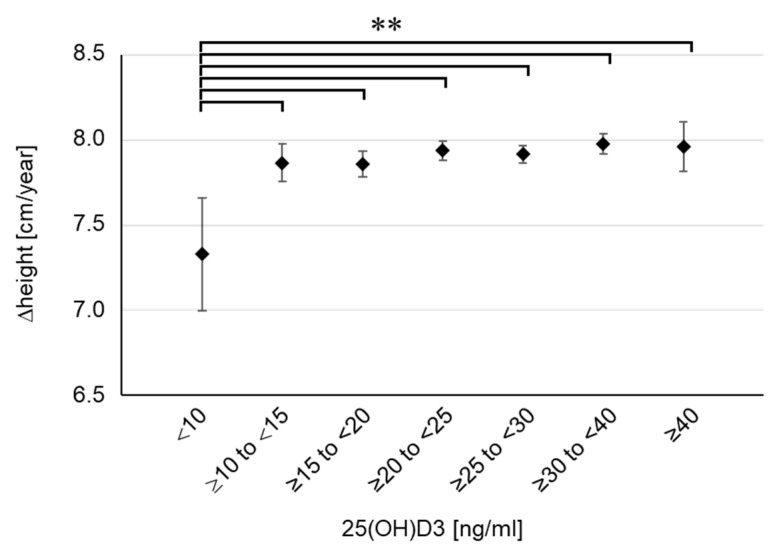
Height growth for each group divided by serum 25(OH)D3 concentration. Δheight means the height growth per year (12 months). Data are shown as mean and 95% CI. ** *p* < 0.01.

**Figure 4 nutrients-14-03325-f004:**
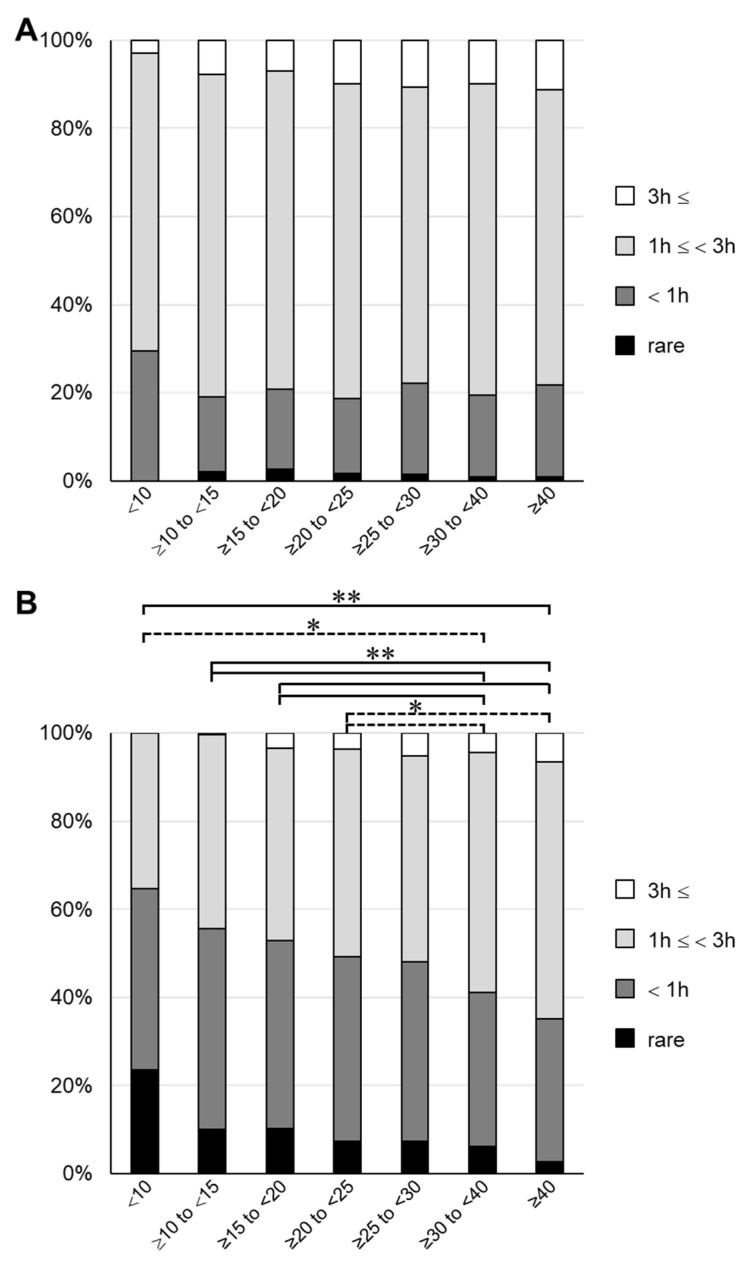
Outdoor playing time for each group divided by serum 25(OH)D3 concentration 3 h:3 h; 1 h:1 h; rare: answer indicating that they rarely play outside. (**A**) Summer. (**B**) Winter. Dashed line, * *p* < 0.05; solid line, ** *p* < 0.01.

**Table 1 nutrients-14-03325-t001:** Profile of each subgroup by serum 25(OH)D3 concentrations.

	Subgroup by Serum 25(OH)D3 Concentrations						
25(OH)D3 [ng/mL]	<10	≥10 to <15	≥15 to <20	≥20 to <25	≥25 to <30	≥30 to <40	≥40
···	(*n* = 40)	(*n* = 254)	(*n* = 524)	(*n* = 913)	(*n* = 925)	(*n* = 846)	(*n* = 122)
Sex	···						
Male (%)	20 (50.0)	122 (48.0)	243 (46.4)	425 (46.5)	488 (52.8)	458 (54.1)	72 (59.0)
Female (%)	20 (50.0)	132 (52.0)	281 (53.6)	488 (53.5)	437 (47.2)	388 (45.9)	50 (41.0)
Birth information							
pregnancy period [weeks]	39.7	39.4	39.5	39.5	39.5	39.6	39.5
birth weight [g]	3093	3066	3095	3073	3073	3089	3068
(95% CI)	(2991, 3195)	(3019, 3113)	(3062, 3129)	(3049, 3098)	(3048, 3097)	(3064, 3113)	(2998, 3139)
birth height [cm]	49.26	49.17	49.26	49.11	49.10	49.26	49.10
(95% CI)	(48.73, 49.79)	(48.92, 49.42)	(49.10, 49.43)	(48.99, 49.24)	(48.98, 49.22)	(49.12, 49.39)	(48.74, 49.46)
At 2 years of age							
height [cm]	83.49	83.78	83.89	83.96	83.86	84.07	84.34
(95% CI)	(82.28, 84.70)	(83.40, 84.16)	(83.65, 84.13)	(83.78, 84.14)	(83.67, 84.06)	(83.88, 84.27)	(83.83, 84.85)
height SDS	−0.55	−0.33	−0.31	−0.31	−0.37	−0.30	−0.25
(95% CI)	(−0.92, −0.18)	(−0.46, −0.20)	(−0.39, −0.23)	(−0.37, −0.24)	(−0.43, −0.31)	(−0.37, −0.24)	(−0.43, −0.07)
weight [kg]	11.47	11.48	11.5	11.46	11.49	11.65	11.57
(95% CI)	(11.00, 11.93)	(11.33, 11.62)	(11.40, 11.60)	(11.39, 11.54)	(11.41, 11.57)	(11.40, 11.90)	(11.36, 11.78)
BMI	16.4	16.32	16.32	16.24	16.3	16.46	16.24
(95% CI)	(16.00, 16.81)	(16.19, 16.46)	(16.22, 16.42)	(16.17, 16.32)	(16.23, 16.37)	(16.11, 16.81)	(16.03, 16.45)
At 4 years of age							
height [cm]	98.5	99.62	99.75	99.87	99.77	100.09	100.26
(95% CI)	(96.99, 100.00)	(99.14, 100.11)	(99.42, 100.08)	(99.63, 100.10)	(99.52, 100.01)	(99.85, 100.34)	(99.59, 100.93)
height SDS	−0.5	−0.07	−0.05	−0.01	−0.06	0.02	0.06
(95% CI)	(−0.87, −0.12)	(−0.19, −0.06)	(−0.14, 0.03)	(−0.07, 0.05)	(−0.12, 0.00)	(−0.04, 0.09)	(−0.11, 0.24)
weight [kg]	15.22	15.44	15.53	15.57	15.45	15.5	15.51
(95% CI)	(14.55, 15.89)	(15.23, 15.65)	(15.38, 15.69)	(15.45, 15.69)	(15.34, 15.56)	(15.39, 15.61)	(15.19, 15.83)
BMI	15.62	15.53	15.58	15.58	15.49	15.45	15.4
(95% CI)	(15.28, 15.96)	(15.40, 15.66)	(15.48, 15.68)	(15.50, 15.66)	(15.42, 15.56)	(15.38, 15.52)	(15.18, 15.62)

95% CI: 95% confidence interval, SDS: standard deviation score, BMI: body mass index.

## Data Availability

The data are unsuitable for public deposition due to ethical restrictions and the legal framework in Japan. It is prohibited by the Act on the Protection of Personal Information (Act No. 57 on 30 May 2003, amendment on 9 September 2015) to publicly deposit data containing personal information. The Ethical Guidelines for Medical and Health Research Involving Human Subjects enforced by the Japan Ministry of Education, Culture, Sports, Science and Technology and the Ministry of Health, Labour and Welfare also restrict the open sharing of epidemiological data. All inquiries about access to data should be sent to: jecs-en@nies.go.jp. The person responsible for handling enquiries sent to this e-mail address is Shoji F. Nakayama, JECS Program Office, National Institute for Environmental Studies.

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
