# Peer review of "Impaired Height Growth Associated with Vitamin D Deficiency in Young Children from the Japan Environment and Children’s Study"

_nutrients, 2022, doi:10.3390/nu14163325_

Round 1

Reviewer 1 Report

The paper is well written and presents a good piece of data. I found it very interesting.

The paper is well written and presents a good piece of data. The only suggestion I have is to change the first statement of the introduction. In my opinion 1,25(OH)2D regulates calcium absorption not 25(OH)D.

Author Response

The paper is well written and presents a good piece of data. I found it very interesting.

Response

We thank the reviewer for the appreciation of our novel findings.

The paper is well written and presents a good piece of data. The only suggestion I have is to change the first statement of the introduction. In my opinion 1,25(OH)2D regulates calcium absorption not 25(OH)D.

Response

In accordance with the reviewer’s suggestion, we have corrected the sentence as follows:

“Vitamin D is the precursor of 25-hydroxyvitamin D [25(OH)D], and 25(OH)D is converted to 1a-25-dihydroxyvitamin D, which regulates calcium absorption in the intestine.”

Reviewer 2 Report

1. Since not every reader is familiar with the latitude of Japan compared to Europe and the US, there should be information provided about the length of the "vitamin D" winter (i.e. the length of the period when no vitamin D synthesis can happen even at extensive outdoor activity) in the different regions of Japan. It may be worth the do a re-analysis of the data based on latitude.

2. it may be worth including Figure S1 as main display item.

Author Response

  1. Since not every reader is familiar with the latitude of Japan compared to Europe and the US, there should be information provided about the length of the "vitamin D" winter (i.e. the length of the period when no vitamin D synthesis can happen even at extensive outdoor activity) in the different regions of Japan. It may be worth the do a re-analysis of the data based on latitude.

Response

We thank the reviewer for raising this important point. We would like to examine the data based on latitude in future studies. We also added the following text to the Discussion section:

“Since this study recruited participants from throughout the Japanese archipelago from latitude 26°N to latitude 43°N, latitude may have an effect. However, in the northernmost region of the country, the hours of winter sunshine number about 110 per month, while in the southernmost region they number about 98, which are not so different. Therefore, decreased outdoor activity due to the winter climate, rather than the number of hours of sunshine, may be the main factor in reduced exposure to sunlight at each latitude.”

  1. it may be worth including Figure S1 as main display item.

Response

In accordance with the reviewer’s suggestion, we have converted Figure S1 to main Figure 2.

Reviewer 3 Report

This a large epidemiological study on the effects of vitamin D status ,in the growth development in children of 2 and 4 years old. The study could comprise a sound epidemiological basis for the effects of vitamin D in childhood,however,there are several parameters that are crucial and have not been incorporated in the current analysis. Offspring height is not only dependent from vitamin D status ,rather from many additional parameters including maternal and paternal height,social conditions, body anthropometry,nutrition,physical activity. The study needs several additional data regarding the aboce fields through validated questionaires and subsequent adjustments for vitamin D findings,as well as at least IGF-1 measurements as a crude estimate of child growth. 

It becomes evident that this kind of analysis,provided here is not of sufficient quality and depth to assume a cause and effect relationship as the authors state.

-Language editing is required

- CVs of the assays used should be reported

-Are there data on maternal-neonatal 25(OH)D at birth? Such interactions could be interesting

Author Response

This a large epidemiological study on the effects of vitamin D status ,in the growth development in children of 2 and 4 years old. The study could comprise a sound epidemiological basis for the effects of vitamin D in childhood,however,there are several parameters that are crucial and have not been incorporated in the current analysis. Offspring height is not only dependent from vitamin D status ,rather from many additional parameters including maternal and paternal height,social conditions, body anthropometry,nutrition,physical activity. The study needs several additional data regarding the aboce fields through validated questionaires and subsequent adjustments for vitamin D findings, as well as at least IGF-1 measurements as a crude estimate of child growth.

It becomes evident that this kind of analysis,provided here is not of sufficient quality and depth to assume a cause and effect relationship as the authors state.

Response

We thank the reviewer for the appreciation of our novel findings, as well as the provision of critical advice. As we state in the limitation section, the current study did not include many parameters pointed out in the comment. However, there are no clear differences in height at birth and at 2 years old among the subgroups by 25(OH)D3 level (Table 1). In addition, BMI levels of the subgroups were almost the same, indicating that their nutritional states were also similar. Therefore, we considered that the background parameters associated with child height, except for 25(OH)D3, hardly differed among the subgroups. The current study may not provide robust evidence to prove the causal relationship, but our large epidemiological analysis could produce new evidence to promote research on the use of vitamin D for child health. We would like to address the suggested issues in future studies.

-Language editing is required

Response

Language editing of this revised version of the manuscript was again performed by Edanz.

- CVs of the assays used should be reported

Response

We revised the sentence in the Method section as follows:

“Serum 25(OH)D3 was measured as one of the variables in the blood samples from the medical examinations by liquid chromatography-tandem mass spectrometry (LSI Me-dience Corporation, Tokyo, Japan).”

-Are there data on maternal-neonatal 25(OH)D at birth? Such interactions could be interesting

Response

We agree that maternal–neonatal 25(OH)D interactions could be interesting. However, we did not examine such data at birth.

Round 2

Reviewer 2 Report

none

Author Response

Reviewer 2:

none

Response

We thank the reviewer for the appreciation of our study.

Reviewer 3 Report

 Please refer in detail to the CVs (inter and intra-assay) of the assays used in the study and just type of analysis (LC-MS/MS)

Author Response

Reviewer 3:

Please refer in detail to the CVs (inter and intra-assay) of the assays used in the study and just type of analysis (LC-MS/MS)

Response

We thank the reviewer for pointing the inadequate description. And, we revised the sentence in the Method section as follows:

“Serum 25(OH)D3 was measured as one of the variables in the blood samples from the medical examinations by liquid chromatography-tandem mass spectrometry (LSI Me-dience Corporation, Tokyo, Japan). The system was calibrated with 6PLUS1 Multilevel Serum Calibrator Set 25-OH-Vitamin D3/D2. The sample testing was validated through Accuracy-Based Vitamin D Survey (ABVD). The intra-assay CV for 25(OH)D3 was L: 9.9% (17.2ng/mL) and H: 9.1% (44.2ng/mL). The inter-assay CV for 25(OH)D3 was L: 4.7% (15.7ng/mL) and H: 5.2% (39.3ng/mL).”